# Perceptions and Factors Influencing Eating Behaviours and Physical Function in Community-Dwelling Ethnically Diverse Older Adults: A Longitudinal Qualitative Study

**DOI:** 10.3390/nu11061224

**Published:** 2019-05-29

**Authors:** Evans A. Asamane, Carolyn A. Greig, Justin A. Aunger, Janice L. Thompson

**Affiliations:** 1School of Sports, Exercise and Rehabilitation Sciences, College of Life and Environmental Sciences, University of Birmingham, Birmingham B15 2TT, UK; c.a.greig@bham.ac.uk (C.A.G.); J.Aunger@bham.ac.uk (J.A.A.); J.Thompson.1@bham.ac.uk (J.L.T.); 2MRC-Arthritis Research UK Centre for Musculoskeletal Ageing Research, University of Birmingham, Birmingham B15 2TT, UK; 3NIHR Birmingham Biomedical Research Centre, University Hospitals Birmingham NHS Foundation Trust and University of Birmingham, Birmingham B15 2TT, UK

**Keywords:** Super-diversity, cultural, ethnic minorities, diversity, social networks, healthy eating, physical function, older adults

## Abstract

Ethnic minorities have a high prevalence of non-communicable diseases relating to unhealthy lifestyle practices. Several factors have been identified as influencing unhealthy lifestyle practices among this population; however, there is little evidence about how these factors differ among a heterogeneous sample living in a super-diverse city. This study aimed to: (1) identify and compare factors influencing eating behaviours and physical function among ethnic older minorities living in Birmingham, United Kingdom; and (2) understand how these factors and their association with healthy eating and physical function changed over 8-months. An in-depth interviewing approach was used at baseline (*n* = 92) and after 8-months (*n* = 81). Interviews were transcribed verbatim and analysed using directed content analysis. Healthy eating was viewed as more important than, and unrelated to, physical function. Personal, social and cultural/environmental factors were identified as the main factors influencing eating behaviours and physical function, which differed by ethnicity, age, and sex. At 8-month interviews, more men than women reported adverse changes. The study provides unique and useful insights regarding perceived eating behaviours and physical function in a relatively large and diverse sample of older adults that can be used to design new, and adapt existing, culturally-tailored community interventions to support healthy ageing.

## 1. Introduction

The population in the United Kingdom (UK) is ageing and becoming more ethnically diverse [1]. In England and Wales, the number of ethnic minority older adults aged 50 years and over has increased from 1.3 million in 2001 to 2.4 million in 2016, and is projected to reach 7.4 million by 2051 [2]. It is also estimated that ethnic minorities will comprise one-third of the total UK population by 2050 [1,3]. However, many ethnic minority groups live in areas of higher deprivation and experience disproportionate health inequalities, predisposing them to a lower quality of life and poorer health than the White British population [4,5,6].

The limited published evidence examining dietary intake in this population suggests that the diets of ethnic minorities who immigrated to the UK are less healthy than when living in their country of origin, due to acculturation [7,8]. The inclusion of diets high in saturated fat and sugar such as pizza, pasties, cakes, sweets, French fries and potato chips as reported elsewhere [9] contributes to these unhealthy eating practices among ethnic minorities. Also, there is reported high prevalence of objectively measured sedentary behaviour and low levels of physical activity among community-dwelling older ethnic minorities in the UK [10,11,12]. For example, a recent study found that, in a sample of 76 community-dwelling older ethnic minority women with a mean age of 70.8 years, only 15% met weekly physical activity recommendations, while more than half were sedentary during waking hours [11]. Furthermore, the prevalence of low physical activity in the UK, defined as the participation in less than one 30-minute moderate/vigorous activity session a week on the average was observed to be high among ethnic minorities [6]. For instance, as compared with 32% in men and 37% women in the general population, low activity levels were found in 51% in both Bangladeshi and Pakistani men, and 68% in Bangladeshi and 52% in Pakistani women [6]. Together with the low physical activity levels, ethnic minorities have an increased prevalence of obesity as compared to the general population; 25% of Caribbean men are obese as compared to 23% in the general population, and the risk ratio for obesity in African women, Caribbean women, and Pakistani women stands at 2.0, 1.43 and 1.48 respectively, as compared to the general population [6]. Unhealthy dietary behaviours coupled with the low physical activity levels within this population are contributors to their increased prevalence of obesity, type 2 diabetes and other non-communicable diseases as compared to the general population [7,13,14]. 

This has significant social and economic implications for the nation, including increases in health and social care costs, and the costs to individuals, families and communities [15]. As such, it is necessary to understand and develop specific policies and interventions that will support community-dwelling ethnic minority older adults to adopt lifestyle behaviours that support them to age more healthily. However, because there are limited published data about the factors influencing healthy lifestyle choices among community-dwelling ethnic minority older adults, it is difficult to develop and deliver effective culturally-sensitive community interventions. Evidence suggests that healthy lifestyle interventions that are culturally/ethnically tailored and that incorporate strategies to optimise healthy eating and physical activity have been shown to achieve significant improvements in lifestyle behaviours, thus contributing to a reduction of the prevalence of non-communicable diseases [16,17,18].

To develop culturally tailored and appropriate interventions, it is imperative to have an in-depth understanding of how the factors influencing dietary behaviours and physical function may differ and interact in this population. There is some published evidence contributing to our understanding of these factors among ethnic minority communities. For example, a study exploring eating behaviours among 36 overweight and obese South Asian men (18–64 years) found cultural factors, lack of time and motivation, and family and other social networks as influencing eating behaviours [19]. Similarly, a study of Ghanaian immigrants aged 25 years and older in the UK found that beliefs, perceptions, accessibility and affordability of food, social networks, acculturation and cultural factors influenced eating behaviours [20]. In addition, an investigation of eating behaviours among young Turkish and Moroccan immigrants in the Netherlands found that religion, identity, hospitality and lifestyle changes were key influences [21]. These studies had samples that were, on average, young to middle-aged, and were in some cases studied as single ethnic groups, which makes it difficult to comprehensively compare and understand how these factors differ among a heterogeneous older ethnic minority population living in a super-diverse community. Furthermore, the cross-sectional nature of these studies does not provide evidence as to if and how these influences may change over time. As such, the present study is intended to fill these gaps in the literature. 

The present study uses a longitudinal qualitative design to: (1) identify and compare factors influencing eating behaviours and physical function among ethnic older minorities living in Birmingham, United Kingdom; and (2) understand how these factors and their association with healthy eating and physical function changed over 8 months. In addition to building on the existing literature, the findings provide insights and novel information to aid stakeholders, policy makers and health professionals in understanding, designing and implementing effective culturally tailored interventions to support healthy ageing in community-dwelling ethnically diverse older adults. 

## 2. Materials and Methods 

### 2.1. Study Design and Setting

A longitudinal qualitative approach was used to gather relevant data. This design was effective in generating a rich dataset that supported an in-depth examination of lifestyle changes that occurred in the target study sample over eight months, the factors that may account for these changes, and the perceived impact of these changes on eating behaviours and physical function [22]. The guidelines from the consolidated criteria for reporting qualitative studies were used to guide this study [23].

The research was conducted in the wider Birmingham area, West Midlands, UK. In this geographic area, ethnic minorities (non-white) comprise 46.9% of the population, with the largest being South Asians (22.6%). The population of ethnic minorities increased in this region by 12.4% from 2001 to 2011, and with the current projected population growth of 7.2% by 2026, the non-white population is expected to make up more than half of the entire population by 2021 [24,25].

### 2.2. Participants and Recruitment

One hundred participants were recruited using purposive sampling coupled with chain referrals and word of mouth. Inclusion criteria were an older adult aged 60 years or older living within the Birmingham area who self-identified as African, Indian, Pakistani, Bangladeshi or Caribbean. Participants were recruited through older age societies, community centres, faith centres and various social events for ethnically diverse older adults across the Birmingham area. Recruitment excluded older adults with a diagnosis of dementia or any cognitive disabilities that might affect their participation in the study. Individuals residing outside of the Birmingham area, institutionalised or hospitalised were also excluded. Maximum variation sampling was employed during the recruitment process to ensure wide representation across older age groups, ethnicities, sexes, religions/faiths and socio-economic groups [26,27,28]. 

Several techniques were employed to increase recruitment, obtain quality data and to increase retention in the study. The first author (EAA) established initial contact with community leaders and potential participants before the start of the study. EAA attended church services and other faith centres’ public events to establish rapport with leaders and potential participants. This enhanced trust between EAA and participants. Community leaders who met the inclusion criteria were also encouraged to take part in the study. Acting as role models, this built trust and enhanced credibility of the study [29]. This strategy was used to optimise participants’ commitment and retention, which in turn enhanced data quality.

The study received full ethical approval from the University of Birmingham Science, Technology, Engineering, and Mathematics (STEM) Research Ethics Committee (ERN_17_1364). All participants provided written informed consent before data were collected.

### 2.3. Theoretical Framework

A constructivist paradigm, with a phenomenological philosophical underpinning, guided this research. As such, the researchers shared an understanding that knowledge is not comprehensively generated without the researcher and the participant [30,31]. This approach enabled the team to set aside perceived assumptions, personal experiences and other biases, to allow for a comprehensive understanding of the factors influencing eating behaviours and physical function from each participant’s perspective. Furthermore, this approach enabled an in-depth exploration of the narrative of factors influencing eating behaviours and physical function, and how these factors changed over time.

Given the aim of this study, an ecological model was used to explore and conceptualise the different levels in which factors influencing eating behaviours and physical function operate [32]. This model best fits this study as it examines multiple factors at different levels influencing eating behaviours and physical function; from the personal level, the social level, the level of the immediate environment and lastly the level of the macro environmental impacts of factors such as of policies and interventions [32]. In addition to these stages, the Bronfenbrenner’s concept of chronosystem was added as the last level of the ecological model to conceptualise the changes, and interactions, of the different levels of influence over time [33] [Adapted model of the ecological model is attached as Appendix A].

### 2.4. Interviews

The conceptual framework coupled with the use of relevant literature was used to design the semi-structured guide for the interviews [10,12,34]. The semi-structured interview guide was divided into two main areas:Eating behaviours and factors influencing eating behaviours.Physical function and factors influencing physical function.This interview guide was iteratively revised at follow-up to include: Changes in factors influencing eating behaviours and physical function over the previous eight months.

Using this interview guide, two in-depth qualitative interviews (baseline and 8-month follow-up) were conducted. The use of this approach was most appropriate as it allowed us to guide the interviews towards important areas that would yield relevant responses [35]. Additionally, to enhance adequate participation and trust of participants during the interviewing process, the responsive interviewing technique was adopted [36]. This technique is flexible and builds a relationship of trust, leading to a give and take style of conversation. Trained translators were employed in instances where the participant had limited ability to communicate in English (*n* = 19). Interviews were conducted from March 2017 to February 2019 at locations of preference for participants including their home, community centres, or at the University. All interviews were digitally recorded and lasted from 25–110 minutes. 

### 2.5. Data Analysis

Interviews were transcribed verbatim during the data collection process. This allowed for the reorganisation of some parts of the interview guide to explore in-depth areas of interest [37]. After transcription, the transcripts were uploaded to QSR NVivo 12 Plus software [38] for reorganising and coding. Data analysis was conducted using directed content analysis [39,40] due to its flexibility of use and its ability to explore a detailed analysis of factors influencing eating behaviours and physical function within this population. Moreover, the use of this approach is well-suited for studies in which there is pre-existing research on the phenomenon being tested [40].

The directed content analysis approach involved preparation, organization and reporting phases. The preparation phase involved full immersion into the data, where transcripts and field notes were read and reviewed to understand the pattern of the data. During the organisation phase, two categorisation matrices were developed: one regarding factors influencing eating behaviours and the other regarding factors influencing physical function. A complete matrix developed on eating behaviours is shown in Figure 1. The complete matrix on physical function is shown in Appendix A. These coding matrices were initially developed deductively using insights from a conceptual framework [32] and existing relevant literature [10,11,12,20,41]. Using an iterative process, these matrices (mainly the generic category and sub-category) were further restructured inductively based on the emerging findings (data-driven) [39].

To ensure rigour, trustworthiness and the ease of use of the matrices, two researchers (JAA and JLT) independently pre-tested the matrices using preliminary data [39]. Afterwards, all discussions around the difficulty of use of the matrices, interpretations of the categories and the data coded were resolved by consensus, and the matrices further restructured. Following this step, the remaining transcripts were coded by EAA using the updated categorisation matrices. During the final step, or the reporting phase of the analysis process, all findings were systematically presented, showing the association between the various categories and sub-categories of the matrices. Field notes taken during data collection were used to support the interpretation of the data. The data were subjected to additional analysis. The use of universalising and differentiating comparative analysis approaches made it possible to understand the similarities and differences between responses by socio-demographic characteristics and any variable of interest in each category [42]. This enabled us to build a logical conclusion of data and to interpret the data more clearly.

## 3. Results

Table 1 presents the demographic profile of the respondents. Out of the 100 participants recruited, 92 participants were interviewed at baseline and 81 at follow-up. Participant loss to follow-up was due to sickness (*n* =4); loss of interest (*n* = 2); and unable to trace participant to schedule the follow-up interview (*n* = 5). At baseline, the mean age (SD) of participants was 70.1 (8.1) with 24% living alone. The majority of the sample comprised of Caribbean and Africans (60%), followed by South Asians (34%) and the smallest proportion were participants identifying as ‘others’: a mixture of the two major ethnic minority groups (6%). Over half of the sample had completed at least secondary school education (58%), while participants with no formal education made up 15.2% of the sample. 

The findings that emerged from the data are presented in detail under the categories below: The differing perceptions of healthy eating and physical function.The personal, social and cultural/environmental factors influencing eating behaviours and physical function and how these factors differ among the sample.Perceived changes to eating behaviours and physical function over the 8-month follow-up period.

### 3.1. The Differing Perceptions of Healthy Eating and Physical Function

#### 3.1.1. Healthy Eating

Perceptions of healthy eating differed by sex, ethnicity, the presence of disease condition(s), religion or age. Among women, less than half (43%) perceived healthy eating as cutting down fat, limiting portion sizes, and healthy cooking methods such as reducing the amount of salt and oil during cooking or opting for steaming and baking as opposed to frying. Similarly, a third of men (34%) shared all these views except perceptions around cooking methods. Across both sexes, home cooked meals were commonly considered to be healthier compared with eating out.

“I think for me, less fat in your diet and I’m thinking about my health as well. Less fat, things that are not fried and smothered in fat when cooking, and a little bit more homemade because then you know exactly what’s going into them is better” (P51, 62 years, Caribbean, Female).

Among participants with diabetes, one-fifth (20%) reported healthy eating as cutting down on amounts of food eaten, skipping snacks and avoiding sugary or fatty fats to manage their health condition. Some (12%) also reported fruits as foods representing healthy eating, as they were told by their health professionals to have enough fruits daily to manage their blood sugar. 

“... I don’t eat…between meals, which is good. So, when I get it, at the right time, I eat and eat everything at a go to help me with my sugar level, to control my sugar” (P30, 60 years, African, Male).

Religious teachings, mainly among Muslim participants, were reported as guiding their perceptions of healthy eating. This was reported by more than half (51%) of the Muslim participants.

“Healthy eating means if you have hunger that much [demonstrating using his hand], you eat only that much [demonstrating using his hand 3/4] and leave that much hunger there [demonstrating using his hand 1/4]. And this is what Islam teaches. They say have a glass of water with [your] meal…No, I don’t want it, because Islam says you don’t eat with water, is not healthy eating” (P48, 69 years, Pakistani, Male).

Across both sexes, traditional foods were mostly considered as healthy foods and tasty as compared to other foods. However, their views appear to differ by ethnicities. For instance, Indian/Pakistani/Bangladeshi male participants with a University/College degree described their traditional foods as being ‘fatty’, ‘greasy’ and “tasty with too much sugar”. In contrast, Caribbean and African participants, across all levels of educational status, perceived their traditional foods as healthier and lower in fat:

“…I don’t think it’s very healthy, the traditional Indian diet has got too much sugar in it, too much oil, they fry a lot, it all makes it not healthy” (P50, 80 years, Indian, Male).

“That is the pivotal point in the West Indies food. I see it as healthy. It keeps you healthy and even though you might not [be] feeling well like you have the flu or you have the pain and everything, with our food, you are never sick too big to go to the hospital” (P19, 81 years, Caribbean, Female).

Additionally, the satiation properties of food emerged among Pakistani/Indian women as an important factor in considering food as healthy. As one participant described, not feeling hungry even after hours of not eating was considered as being healthy for her. Hence, she considers the traditional foods that could satiate her for longer hours as healthy foods.

“That [traditional foods] is healthy eating…. It’s more filling. So, [I] think it’s healthy to eat that, as opposed to feeling hungrier” (P88, 73 years, Pakistani, Female).

#### 3.1.2. Physical Function

Physical function was largely perceived as participation in exercise and general fitness level. It was also labelled as ‘strength,’ which participants viewed as the ability to carry out activities of daily living without difficulty.

“Physical function means how active you are. How active, how you can be independent. How you can do things on your own. In terms of flexibility, how you can [be] flexible with your body movement, and all of that” (P37, 63 years, African, Male).

The study also explored the description of older adults’ physical function levels compared with their colleagues. Of the majority (53%) that commented on this, two-thirds rated themselves as having a high physical function (at least eight on a 10-point scale). The remaining participants reported low self-rated physical function and attributed it to increasing age, ill-health and pains as illustrated in the quote below.

“I’m maybe below average, I’m not good, but I’m not very poor too, I think with the few aches and pains now I will say I am about three out of 10” (P39, 70 years, Indian, Male).

Like healthy eating, the concept of improving physical function to manage health conditions also emerged.

“If you want to keep mobile you have to keep your fitness [physical function], I tell you everything is about my blood sugar, it [fitness] does affect your blood sugar and stuff like that” (P63, 67 years, Caribbean, Female).

#### 3.1.3. Priority and Association between Healthy Eating and Physical Function

Across educational status, most men (67%) considered healthy eating to be more important than maintaining physical function. This was attributed mainly to the fact that due to age-related decline in physical function, they believed nothing could be done about improving their physical function levels, hence there was no need to prioritise it above healthy eating. 

“To eat well is more important, with my age there is no running round about. I don’t bother with the fitness [physical function] now because with my age, but when my daughter cooks me good food, I can still eat well” (P52, 92 years, Caribbean, Male).

In contrast, a few (12%) women thought otherwise and explained that with changes in physical activity level, they prioritised their physical function over healthy eating, as they believed this would help them to maintain their activity level and a healthy weight.

“I think exercise is more important these days because you’re not doing any physical work, because one time you used to eat and do physical work and that used to drain you out” (P45, 70 years, Caribbean, Female). 

The perceived association between healthy eating and physical function was mixed, in that some participants (34%) believed that they were independent of each other and had no relationship, while a small number of participants (6%) described a relationship between these two factors, stating that healthy eating supports them to maintain a good level of physical function and vice-versa. More women self-identifying as Caribbean or African expressed the latter view as compared to men or women from other ethnic minority groups, as illustrated by the quotes below:

“I think eating well helps your physical function, but sometimes eating well, you still have to do something with the physical function. No point in eating well and just sitting around, you have to do both. They both complement each other” (P75, 62 years, Caribbean, Female).

“Between the physical function and eating? There’s no connection between them, just when you’re tired you eat, I don’t see any link” (P61, 76 years, Bangladeshi, Male).

### 3.2. Factors Influencing Eating Behaviours and Physical Function

Personal, social, and cultural/environmental influences on eating behaviours and physical function were reported. Participants’ responses suggested that these influences differed across ethnicity, sex and socio-economic status. Figure 2 illustrates the main categories and the sub-categories of those identified.

#### 3.2.1. Personal Influences 

Participants highlighted several personal factors influencing their eating behaviours and physical function. As shown in Figure 2, these included current health conditions, pain, body image, age, and retirement status. 

##### Health Conditions

Some participants (24%) reported making changes to their eating behaviours such as changing diet, eating patterns or cooking methods to manage their current health conditions. Others (6%) also reported improving their physical function through walking and gardening to manage their current health conditions. These responses differed by sex and disease severity. For instance, 13% of men reported the influence of health on their diet, stating they avoided sugary snacks and reduced portion sizes. In contrast, women (11% of the total sample) reported changing their cooking methods, substituting their regular (or staple) foods with healthier options, and reducing portion sizes. 

“… since I was diagnosed [with] this type 2 diabetes, I watch what I eat in between meals. So, I don’t take anything sugary…I don’t eat between meals. So, when I get it [food], at the right time, I eat and eat everything at a go to help me” (P37, 63 years, African, Male). 

“…. No, I don’t cook with salt, and I don’t fry again, my cooking has changed, and I don’t eat our [traditional] food sometimes…Well, your blood pressure, isn’t it? I’m on blood pressure tablets, so I need to change. I am also diabetic too, I have to change [my diet] now” (P72, 68 years, Caribbean, Female).

The fear of losing independence and other consequences accompanying complications from diabetes and other diet-related non-communicable diseases emerged as motivating factors for some (12% of the sample) to eat healthily and maintain their physical function. As one participant explained:

“Because I have type 2 diabetes and I’ve seen, [people] actually suffering. I don’t suffer with it, but because I have seen what it can do and has done to members of my family, and I’m not prepared to go there, that’s my impetus, that’s my motivation, that’s my drive. To eat healthily, take a lot of regular physical activity to keep the complications at bay” (P67, 63 years, Caribbean, Male).

##### Pain

Pain resulting from health conditions was mostly described by participants to be inhibiting them from doing activities to maintain their physical function. 

“Also, like I’ve got a bit of a dodgy left knee mainly from my arthritis. Sometimes that hinders me from doing a lot of walking… because of my knee pain, so I just stay put” (P65, 70 years, Caribbean, Female). 

Pain was also reported to be affecting cooking and shopping, as most could not either stand to cook or go about their regular shopping. A man living alone describes how pain from his knee injury is inhibiting him from preparing traditional meals as they require much attention during cooking, so he has resorted to consuming simple, ready-to-eat meals.

“I am not able to have that food [traditional food] as I will like [to]. Now I just have these easy ones (ready-to-eat meals) as you can see.... buying it myself, it takes time to cook, it takes time to prepare curry goat [traditional Caribbean soup], and with my bad knees, I would not be able to stand for it” (P58, 65 years, Caribbean, Male).

##### Body Image

Body image also emerged as a factor influencing eating behaviours and physical function. More women (23%) reported this as compared to males (8%), describing a healthy and ‘desirable’ weight as a means to good health and a way to manage their health conditions. These were reported to influence their portion sizes, types of foods eaten and enjoyment at meal times.

“I don’t eat as much cheese as I used to, I used to love cheese, I used to love crackers, and I very rarely eat that. I mean, you see, another thing, it’s not just eating to sustain me, I’m trying to keep my weight down as well, so there’s that sort of consideration as well” (P65, 70 years, Caribbean, Female).

Maintaining a healthy body weight was also described by some (10%) women participants as a motivator to keep up their physical activities, thus helping them to maintain physical function. 

“If you walk, your joints are fine; you keep the weight, so you keep your body and your muscles healthy, your physical function will not fall” (P74, 85 years, Caribbean, Female).

Interestingly, of the majority (93%) that reported having a diet-related non-communicable disease(s), very few (8%) reported making efforts to improve their physical function or commented on the importance of maintaining a healthy weight. However, the few that stated making efforts to lose weight shared similar difficulties in achieving their weight loss goals. 

##### Age

A third of the participants reported the influence of age on eating behaviours and physical function. Those that recognized a decrease in appetite and decline in physical function attributed it to age.. This participant explains how he is becoming more health conscious with age, hence the need to make certain adjustments for better health.

“I think we are getting more health conscious, put it this way. As we get older, nowadays me and my wife, both think before we eat. In other words, restrict, eat smaller portions” (P14, 60 years, African, Male).

More men living alone than women commonly expressed age-related declines in senses such as taste to be affecting their food intake. As narrated by a man living alone, loss of taste makes eating difficult and less enjoyable.

“I realise after spending a full day to cook food [to feed me] for three days, I eat the first day, and there’s no taste left…so it become[s] difficult to eat” (P14, 60 years, African, Male).

##### Retirement

Retirement was commonly identified as contributing to changes in lifestyle, in that participants had more time for cooking, shopping or doing physical activities. Some (28%) participants stated this had substantially changed their eating behaviours as compared to when they were working:

“I didn’t have much time to watch what I was eating [while working], and I used just to grab things and go because I was all [the] time working, I probably wouldn’t have bothered with green stuff like spinach, I only used to have spinach occasionally. I have it more often now” (P28, 82 years, Caribbean, Female).

In contrast, a few (5%) of recently retired participants thought otherwise. They described the increased ‘free’ time to be causing them to eat more and resulted in them gaining weight as compared to when they were working: 

“Probably my style has changed because I’ve retired. When I used to work, I was going to work with a cup of tea and one toast, and I would run away to work. Now I’m a bit relaxed, so I have a good breakfast, a good lunch and a good dinner and that has put on my weight. I’ve got all the time to eat” (P14, 60 years, African, Male).

Regarding physical function, only a few (15%) retired participants, predominantly women, reported that with the increased amount of time available as compared to their working days, they have time to spend doing activities that could improve their physical function. 

“Yeah, I was sitting from the car to the office, sit down [through] meetings…I think I’ve lost a bit of weight and fitter since I left work. I bought a bike and now [with] my friend who is retired as well, we’d be out, probably get out in the park” (P68, 62 years, Caribbean, Female).

“Yeah, you know, when you go to work, especially Monday to Friday, and then you sit in the car, it limits your physical function a bit because you aren’t able to exercise your body, but now that I am home that is not an excuse. I have more time and I walk around” (P30, 61 years, African, Male).

Contrary to this, four women and a man complained of lack of time after retirement, as they have to either care for their grandchildren or have an abundance of social commitments relating to their faith/religion and other social groups. Among participants still in active work, a few (22%) across both sexes reported their work schedules as a hindrance to undertaking activities that might improve their physical function.

“It [is] either to do with working too late….and then after I’ve worked, I’m so tired, and after I’ve eaten, then it’s too late to get on the cross trainer or do walking…all I can do is just eat and watch TV” (P75, 62 years, Caribbean, Female).

#### 3.2.2. Social Network Influences 

Social network influences included those of family, friends, and the community. Almost all participants described how meal times were more enjoyable when the family size was large. Participants indicated that reduced network size as they have aged, made it challenging to cook often or to cook a variety of foods, due to greater likelihood of food waste. 

“I don’t cook a lot ever since I am on my own, it has changed ever since my youngest daughter left home to go to the university. Sometimes, I will prepare food and when I am supposed to eat, and I don’t want it anymore… Because being on your own I am not looking forward, I am not thinking about food…on my own, I will say ’why will I bother?’. So I will go for a fruit or anything and end up freezing the food…” (P3, 61 years, African, Female).

“I wouldn’t say my eating habits have improved, since the children all left I don’t cook that again, when the children were at home, we used to have a lot of eating times, now it has reduced.....because the excuse is, there’s more to eat when I cook and I don’t wanna waste it” (P24, 61 years, Caribbean, Female).

However, Sunday dinners were described as a special day of the week where children and other family members would visit. Some (30%) said they enjoyed eating and might eat more on these days, as there is a variety of foods available. In contrast, others (6%, all women) reported that the presence or lack of social interactions at meals had no effect on their eating behaviours.

“I tend to put in more effort in what I do and what I cook while if I am on own, I don’t do that much. Yeah, I might just cook something simple but when they come [children] I make it more special...I have more to eat when we are all seated” (P19, 81 years, Caribbean, Female). 

“I don’t really mind. It’s nice to eat with people, but I’m quite content to eat on my own as well. It’s not an issue whether people are there or not, I eat [the] same” (P75, 62 years, Caribbean, Female). 

It also emerged that a few (17%) men preferred the act of commensality, and attributed its importance to the fact that their spouses and children supported them to eat healthily, especially when eating together during the meal:

“Yeah, eating with them (family) impacts on my eating, it does, because you can see my wife, she’s not a very big eater, and she’s very conscious of what kind of foods she puts in her body whereas sometimes I’m not conscious. Eating with her I tend to eat healthier, I do, yeah” (P60, 61 years, Caribbean, Male).

Some participants living alone reported that they had regular support with shopping and cooking from their children or friends. They further explained that with a decline in physical function, this was necessary as they could not shop or prepare their meals. 

“She [daughter] does the cooking and [brings] it over to me...I do breakfast myself. But the cooking I can’t stand that long to do it…my daughter does the shopping once a month for me” (P29, 80 years, Caribbean, Female).

Regarding physical function, a few (11%) participants expressed the support and encouragement received from family and friends as having a positive impact on their physical function. Instrumental support, such as taking them to the gym, paying for personal trainers, buying gym equipment, or providing them with information on how to improve their physical function was highlighted as beneficial. Also, participants that attended community/age-well society meetings considered this engagement as beneficial to maintaining their physical function. At these meetings, some of the activities included leisure activities, board games and occasionally exercise and nutrition classes. 

“…So, when I joined here, it has help[ed] me [to] socialise, and the group exercise is keeping me active, my [physical] function is improving” (P15, 75 years, Pakistani, Male). 

#### 3.2.3. Cultural and Environmental Factors 

Across all ethnicities and age groups, the influence of various cultural (traditional) food practices such as eating certain types of food and the food preparation were commonly expressed by participants. Participants referred to these as their “identity”, “heritage” or “culture” that were important in bringing back memories and providing a sense of comfort and ‘proximity’ to their home country.

“I will have to eat Caribbean foods, [it] makes you feel good, and makes you feel as if you are home, [and] that’s part of you. Potatoes are not part of me, chapattis are not part of me. So, I’ve got this [Caribbean foods] as part of me…. Sometimes when you eat it, it brings back memories of certain things, so that’s why I do enjoy it when I cook it. I’ll cook it to remember a lot of things” (P83, 71 years, Caribbean, Female).

It emerged that some participants did not like the format and content of some health education messages relating to their traditional foods. While messages around reducing portion sizes were regarded as feasible, messages involving the removal of their traditional food were heavily criticised. For instance, eight participants (six males and two females) described dietary advice that meant they should stop eating their traditional food completely as not realistic or feasible:

“A dietician once told a black woman from the Caribbean to cut out her Caribbean food altogether. You never tell a white person to cut out their potatoes. To deny someone their culture and heritage will not work for me…food is what makes a person what they are….” (P67, 63 years, Caribbean, Male).

The influence of the multi-cultural and super-diverse environment of Birmingham on access and affordability of food was also raised by participants. In general, participants highlighted the diversity in Birmingham as having an important role in ensuring accessibility to, and affordability of, some of their traditional foods which was not the case in cities/areas with less diversity. Besides accessibility and affordability, diversity was also reported as helping them to feel comfortable when buying their traditional foods, as one participant recounted: 

“It [diversity] has made myself very comfortable, I feel comfortable because I know they are other people also having the same sort of food. At the market, I see people buying food like me. It makes me feel happy and [at] one with the place” (P6, 62 years, African, Male).

“In Birmingham, you can get anything you want as far as African food is concerned. You can get anything you want here. I think it has made us eat well as we can get yam, gari [flour grains from cassava] or anything at any time” (P97, 72 years, African, Male).

As revealed above, participants considered themselves to be eating ‘healthily’ because they can access the foods they prefer and value. Also, it was revealed that the diversity in the Birmingham area had encouraged the major grocery chains to sell some traditional foods, thus making it more accessible. Probing further, it emerged that most women, living in the most deprived neighbourhoods per the UK Index of Multiple Deprivation (IMD) scale, described the prices of traditional foods in these supermarkets as slightly more expensive compared with those sold in their local corner shops or the budget market located in the city centre.

“Because of the diversity, you can now shop for traditional foods at the Asian shops, at Tesco, at Lidl, at Asda, but it’s a bit dear as compared to the market at Bullring [budget market in the city centre]” (P63, 67 years, Caribbean, Female).

Predominately, more African and Caribbean participants describe their traditional foods as expensive as compared to South Asian participants. These responses were common among both sexes and across the IMD quartiles. However, the influence of traditional food prices on food choices was mixed among this sub-population. While a few (11%) reported that it had no effect, more participants (32%) thought otherwise. As shown in the contrasting quotes below:

“Well, another thing again… provided it’s good for you; I never think it’s too expensive. I never question that. It doesn’t matter…. If I’ve seen it, want it, I’ll buy it” (P96, 76 years, African, Male).

“I don’t eat much of the Caribbean [food], we eat more rice and the other stuff because the Caribbean food is very expensive. Rice is cheaper as compared to green bananas, yams, and sweet potatoes…. Yes, I do enjoy these more, but we just have to manage with that rice, because of the prices” (P20, 86 years, Male, Caribbean).

Exploring the importance of diversity further, a few (6%) of the African and Caribbean women reported that the multicultural nature of Birmingham provides them with a platform to share healthy practices across ethnicities. For instance, they reported sharing best practices with friends and church members on healthy ingredients and other efficient ways to manage and control chronic diseases with food.

“Very multicultural…. because especially with the Sikh community, because they tell me about what things are good. My friend, she’ll tell you how to use these powders. So, I think the multicultural [environment], you pick up and share different ideas” (P68, 60 years, Caribbean, Female).

Finally, some (32%) participants highlighted the effect of bad weather, notably increased snow and rain, as making it unsafe and demotivating to participate in usual outdoor activities such as walking or gardening. In general, the cold weather was reported as the main cause of an increase in joint pain, making it more difficult to engage in physical activities.

“The only thing that would prevent me from going out is the weather. If the day is a normal day, I’ll visit a friend. Doing something to the car or something like that, keeping fit…but if it’s raining or snowing, my knees start to hurt, I just stay in bed” (P66, 73 years, Caribbean, Male).

Besides the influence of the weather on maintaining physical function, it also emerged that with the cold weather, eating patterns are disrupted as narrated by one participant, as she prefers vegetables in the summer as compared to the winter:

“Like with my vegetables, in winter I really don’t like to eat vegetables, I don’t know why. In the summer I will eat any amount of vegetables, but when it comes to winter, my mind just blocks vegetables out. I don’t know why” (P72, 68 years, Caribbean, Female).

### 3.3. Perceived Changes at Follow-Up

At the 8-month follow-up, one-third of the participants reported significant changes in their eating behaviours and physical function. The rest of the participants reported little or no changes. Those reporting significant changes described recent sickness and hospitalisation, falls/accidents, fasting, and changes in the level of support as factors influencing their eating behaviours and physical function. The impact of these changes was mixed, while some (42%) described the changes as positive, that is supporting them to either eat well or improve their physical function, the majority (58%) considered these changes as negative, thus affecting their physical function or eating behaviours. Predominantly, more men reported adverse changes compared with women. 

“Well, it [eating behaviours] has changed quite a lot, to be honest. Because as I said, I used to have a wife…my wife used to cook, but now I have to do all that myself and it is not easy. I don’t think I am eating that well” (P58, 65 years, Caribbean, Male).

Most men who reported these negative changes were living in the most deprived neighbourhoods as per the UK IMD scores. However, this was not the same trend with most women reporting positive changes, who lived in areas of lower deprivation. There were also differences in responses by age among the men reporting changes. Men who were older than 70 years expressed more negative changes as compared to those aged less than 70 years. However, this trend of responses again did not apply to women, as their responses to experiencing changes were evenly distributed across the age range. Participants who engaged in health education opportunities during the study described these as accounting for their experiencing positive changes over the 8 months. A woman reporting a positive change, explained how her recent visit to a clinic-based health education session during the study period led her to make healthy dietary changes.

“To be fair, I have cut out sugar. That’s one of the things I’ve done since I saw you, I did go down from one [teaspoon of sugar] to half. Now, I can drink things. Well, I went to the hospital and the nurse said a group of us who are pre-diabetic, that it will help….” (P68, 60 years, Caribbean, Female). 

The introduction of fasting (both religious and non-religious) was also an important change discussed by participants, especially those reporting sicknesses during the follow-up period. As described by one participant, introducing fasting was perceived as an effective way to lose weight and regain fitness:

“I’m putting on the weight, because of the drugs or because I was in the hospital or whatever the reason might be. So, now I decided to start fasting. If I’m fasting, I feel lighter with my weight and more energy” (P80, 84 years, Indian, Male). 

Consistent with the baseline responses, changes in weather were repeated as influencing physical function. Participants interviewed during the summer months at follow-up expressed this as an enabling factor to maintaining their physical function and to getting out of the house, engaging in activities, visiting community clubs and religious groups. In contrast, participants interviewed during the winter perceived it as a barrier to maintaining their physical function as they opted to stay in bed at most times. 

“The weather, I will say yes. You see, like, when the weather [is] hot and nice [like] now, it makes you want to come out of the house, and find different things to do but when you [researcher] came in the winter, I was always in bed, you don’t want to go out, no, there is no motivation to do anything” (P72, 68 years, Caribbean, Female). 

## 4. Discussion

This study found diverse perceptions of healthy eating and physical function among ethnic minority older adults that were in some cases shaped by socio-demographic characteristics. The findings revealed that eating behaviours and physical function were influenced by a wide range of factors, from personal, social to environmental factors. At follow-up, more men reported negative changes to eating behaviours and physical function. The key findings are summarized and discussed as follows. 

### 4.1. Perceptions of Healthy Eating and Physical Function

Perceptions of healthy eating differed by sex, ethnicity, age and educational attainment. Common perceptions of healthy eating were a reduction in portion sizes, cutting out fat and not skipping meals. In addition to these, women perceived healthy eating to be related to cooking methods. Some of these findings are consistent with previous studies conducted among ethnic minorities and the white English population [43,44,45]. However, the comparison of perceptions by socio-demographic characteristics within a large sample of ethnic minority groups makes the findings of this study unique.

One of the significant findings is that South Asian women considered healthy eating to be eating more satiating foods. In previous research, South Asian participants preferred traditional foods for their satiating properties, however, they added that these were not healthy [8]. The study also found that participants with a medical condition and having recently attended a relevant health education session were likely to report making changes to their diets or improving their physical function. Similar to the Hertfordshire study, older white British participants commented on diets during their upbringing and the effort made to change later in life due to health reasons, citing the influence of the media [46]. The reliance on the media was different in the current study. Participants in this study relied mainly on health care providers for information. This could be attributed to the relatively lower literacy rates among ethnic minorities [47], or alternatively, to the fact that most media sources might not have culturally-tailored messages targeting these populations. 

Differing perceptions about traditional foods were present between ethnicities. Those self-identifying as Indian or Pakistani perceived their foods to be generally unhealthy, whereas African and Caribbean older adults viewed it otherwise. These findings are in contrast to a comparative study of African and Caribbean women in the USA, in which African women were more likely to consider their diets as unhealthy and hence made changes to foods and cooking methods [48]. Consistent with this study, Caribbean women valued their traditional foods and cooking practices and believed it kept them healthy [48]. The perception of Indian and Pakistani older adults of the healthiness of their traditional foods within this study also resonates with the findings of a study looking at perceptions of British-Pakistani older adults living in Bradford, which found that they considered their traditional foods, among other things, to be heavy and unhealthy [8]. Furthermore, African and Caribbean older adults in the present study also considered their traditional foods expensive compared with their South Asian counterparts, and therefore had to make uncomfortable dietary adjustments in some instances. This might have implications for the enjoyment of meals and diet quality and should be investigated further in future studies.

Most participants rated their physical function highly and perceived that physical function was related to health/fitness, exercise, and the ability to carry out activities of daily living. The present study found that participants’ self-rated health was in most cases opposite to self-rated physical function. Previous research reported that the social and economic disadvantages experienced over the life course partly accounts for low-self reported health [49], and that low self-rated health was associated with limitations in activities of daily living [50]. In the present study, participants commonly expressed a fear of losing independence, which could have influenced the discrepancy between self-rated physical function and self-rated health. Some recounted stories of family or friends losing their independence and having a poor quality of life; this fear of loss of independence may have been a reason for the high self-rated physical function, and a possible motivator to maintain physical function as explained in the extended Health Belief Model (HBM) [51]. 

### 4.2. Factors Influencing Eating Behaviours and Physical Function

The present study found that at the personal level, health conditions, pain, age and retirement were factors influencing eating behaviours and physical function. At the social level, changes in social support, living arrangement and community involvement were identified as key factors. At the cultural or environmental level, superdiversity, the cultural identity of food, and the weather were identified. These factors also differed by socio-demographic characteristics of participants and were in some instances consistent with previous studies [19,20,21,34,46,52,53]. However, there were some unique differences compared with the existing literature. For example, at the physical environment level, the influence of super-diversity was quite novel. Super-diversity refers to the unprecedented collection of different nationalities, faiths, languages, cultures and ethnicities in a society [54]. This study found that participants living in super-diverse communities not only appreciated the accessibility and affordability of traditional foods, but also, they described it as providing them with a comfortable environment to eat and shop. This is in contrast to previous studies, where White British older adults expressed shopping difficulties, such as access to supermarkets and being able to purchase single portion foods [46,55]. Possibly, the influx of ethnic minority corner shops and supermarkets due to the diversity of the geographic area might account for the differences found between the present study and previously published research. Our findings also contrast with those reported in a study from the Netherlands, in which young immigrants had difficulty in reconciling lifestyle from their country of origin and the host country [21]. The increased length of time that the older participants had lived in the UK may account for these differences. 

Furthermore, the diverse environment was reported as enabling minority ethnic older adults to share cultural practices. The multicultural nature of these communities was serving as a basket of knowledge for them to share culinary skills and recipes for combating diseases. Thus, it highlights more evidence of heterogeneity among ethnic minority communities. The differences shared among and within these groups could be used in tailoring community-based interventions to give the desired results as proven effective in previous interventions [16,18]. 

At the personal level, most of the findings were consistent with previous literature [41,56,57]. However, the comparative analysis of these factors revealed new insights. For example, not all participants in this study saw retirement as impacting eating habits positively, as data from white British older adults has previously found [46]. Some participants saw retirement as negatively affecting their eating behaviours and physical function, as they either had more time to ‘overeat’ or not get out of the house more to improve on their physical activities. In some cases, increased religious and family commitments after retirement also affected eating behaviours and physical function. 

At the social level, the act of commensality was described as supporting older adults not only to eat better but to enjoy meals more, and this is comparable with previous literature including white British older adults [55,58]. Our data suggest that eating together might also be enhancing the variety of meals, as participants reported eating more varied diets when with company. This concurs with a study of factors influencing varied diets in community-dwelling older adults in Europe [56]. Furthermore, the dietary advice and support received from family and friends during meal times were also considered vital. This could contribute to diet quality as observed in previous studies [46,56], but more investigation is needed in this population to confirm this supposition. In addition to the family’s support in eating better, our data suggest that family support also played a role in keeping some participants active through encouragement, buying exercise equipment and buying gym memberships. This was different from other studies in which family members were overprotective and in some cases prevented older adults from engaging in physical activities due to fears for their safety and health [57]. One possible explanation for this difference could be attributed to the fact that family and friends within this study were reported to be physically active and interested in healthy eating, which may not have been the case in previous studies. 

The regular meetings of ethnic minority older adults at faith centres, clubs and community centres were also reported as significant influences on eating behaviours and physical function. These meeting places were expressed as an avenue to worship, exchange ideas, offer support for colleagues during difficult times and engage in leisure activities with friends. Also, a few of these gatherings offered meals, exercise classes or nutrition talks; these could form part of the ‘active ingredients’ of how social networks influence health outcomes in later life. These findings concur with previous research which found that increased social networks and greater involvement in leisure activities did not only improve diet quality in older adults but also enhanced resilience to cope with lifestyle changes [58,59].

Given that not all older adults experience the same level of changes in eating behaviours and physical function as they age, this study provided novel findings with regards to perceived changes with time and how these differed within this sample of ethnically diverse older adults. Our results indicated that a third of the participants experienced changes within a rapidly short period. Some of the changes during follow-up were the introduction of fasting (with the primary goal of weight loss), changes in the level of support, sickness, accidents/falls and changes in the weather. Men living in higher areas of deprivation were more likely to experience negative changes perceived to be affecting their diet and physical function as compared to women. Previous literature supports these findings as they suggest that men, especially those with poor cooking skills, are likely to have negative changes to diet after losing their spouse, as most meals were previously prepared by their spouse [60,61]. 

Considering that most participants in this study had one or more diet-related non-communicable disease, it was surprising to note that at follow-up only a few participants reported making a positive change to improve their health condition(s). This is contrary to previous findings, where a majority attributed changing their diet due to their health conditions [52,62]. One possible explanation for these differences could be the maintenance of eating behaviours for reasons beyond health concerns among this population. The cultural identity of food appeared to be more of a primary reason for eating habits than health, as participants purported a variety of benefits of eating their traditional foods. 

Perceived negative changes that affected both men and women were the change of weather. Staying at home during these months may not only affect their physical function, but also makes it difficult for participants to engage in other community activities, thus the likelihood of experiencing declines in social ties. This is a commonly-reported issue affecting the ability of adults to engage in regular physical activity across the lifespan [12,63].

### 4.3. Perceived Priority and Association of Healthy Eating and Physical Function

Our study revealed that community-dwelling ethnically diverse older adults are more likely to consider healthy eating and physical function to be separate entities with no relationship. Furthermore, participants commonly prioritised healthy eating over physical function for general health. While more men prioritised healthy eating over physical function, some women prioritised physical function, mostly for weight management reasons. However, more women than men, especially from African/Caribbean backgrounds, viewed healthy eating and physical function as related. This has implications for how they are likely to understand and adopt healthy lifestyle practices based on the already existing generic health education messages. Considering that this population is more sedentary, with some possibly having poorer dietary habits due to acculturation [7,10,11], this justifies the need to build awareness of the interrelation of healthy eating behaviours and physical function. This could serve as a primary factor in increasing knowledge and informing strategies to assist these older adults in making the adoption of healthy lifestyle practices more feasible.

#### 4.3.1. Implications for Policy and Practice

Given the importance of the cultural significance of eating patterns, culturally tailored educational messages about healthy eating using traditional foods as practical examples could be designed by health care providers to ensure proper adherence to dietary advice. Additionally, considering the differing perceptions and influences on eating behaviours and physical function, community health interventions and health messages, when delivered, should consider these heterogeneities between and within ethnicities to avoid broad generalizations and stereotyping, and to celebrate and harness this diversity to develop and deliver programs that can achieve the desired results. 

Some participants were conscious of their health as they aged and ready to choose healthy lifestyle options. Hence, resources highlighting the nutrient content of traditional foods, such as the Ismaili nutritional data on traditional foods [64], could be made readily available. This tool could also be used to equip health care providers to deliver quality dietary advice per the UK recommendations, and could inform the development and dissemination of an ethnic minority version of the Eat-Well plate [65]. Additionally, healthy traditional cooking classes could be more evidence-informed in their design and content, and delivered at social, religious and community centres to support older adults in adopting healthier cooking methods. The addition of these activities to these accessible and ethnicity-specific meeting places would encourage attendance with the potential of improving social networks and reducing the impact of loneliness on eating behaviours and physical function found in this study.

Following health education, participants who reported they had been identified as having pre-diabetes made significant efforts at follow-up to change their diet; thus, increased early screening and dietary advice would be beneficial in this population. Also, given that some of the meeting places for ethnic older minorities are willing to support these activities, health authorities could better deliver assessments and health education at these centres to ensure increased access and acceptability.

#### 4.3.2. Strengths and Limitations

The inclusion of an under-represented population is a major strength of this study [66]. Moreover, the sample size of the present study was relatively large for a qualitative study, which benefited the comparative analysis of ethnicities, sex and other socio-economic characteristics. The study also benefited from a rigorous data analysis process that enhanced the reliability and trustworthiness of the findings. Two experienced researchers independently coded a sub-sample of interview transcripts using the developed categorisation matrices. This could have reduced bias and misinterpretation of data [67]. The longitudinal qualitative design also adds strength to the study.

Additionally, the diverse experiences of the research team were an advantage. JLT and CAG are both senior female researchers with vast experience of working with older adults and ethnically diverse communities. EAA is a doctoral researcher with expertise in qualitative research in diverse populations. As a male researcher, he was able to recruit more men than previous studies. However, we must recognise the potential limitations in how we interpret the data due to the younger age, gender, and different ethnic backgrounds of the research team.

Likewise, the use of interpreters for 19 interviews could have introduced translation bias. We felt it was essential to utilise trained interpreters to allow for the inclusion of participants with low English literacy to ensure the sample was more representative of the current level of literacy in Birmingham [24]. Recognising the potential of compromising validity and reliability of the study due to the use interpreters, we adopted the following strategies to improve the quality of the data [68,69]: (1) interpreters were carefully interviewed and selected, and were trained to ensure they fully understood the interview process and their defined roles; and (2) a second interpreter who understood the respective language of the participant reviewed and evaluated the recording of the first interview conducted with the trained interpreter before any further interviews were conducted. This enabled us to revise the interview guide and work with the interpreter to improve the quality of the translation. These strategies should have contributed to enhancing the reliability and validity of the data. Additionally, eight months of follow-up maybe considered a relatively short period of time to observe particularly environmental changes, hence the authors recommend future studies incorporating a longer follow-up period. Lastly, despite efforts to recruit similar numbers of older adults across the major ethnic minority groups living in the Birmingham area, the sample underrepresents older adults of Bangladeshi origin and South Asian women. Hence, the findings should be interpreted with caution.

## 5. Conclusions

This longitudinal qualitative study found diverse perceptions of healthy eating and physical function among ethnically diverse older adults. Traditional foods were highly regarded as healthy foods by African and Caribbean older adults. The presence of super-diversity was reported as positively influencing accessibility and affordability of traditional foods. Also, diversity supported them to feel comfortable and encouraged to shop, eat and engage with other cultures. More men than women perceived healthy eating and physical function to be unrelated, and prioritised healthy eating over physical function. At follow-up visits, more men than women reported adverse changes such as changes in the level of support (decline in social networks), sickness and accidents/falls. These changes were perceived to negatively affect eating behaviours and physical function. 

## Figures and Tables

**Figure 1 nutrients-11-01224-f001:**
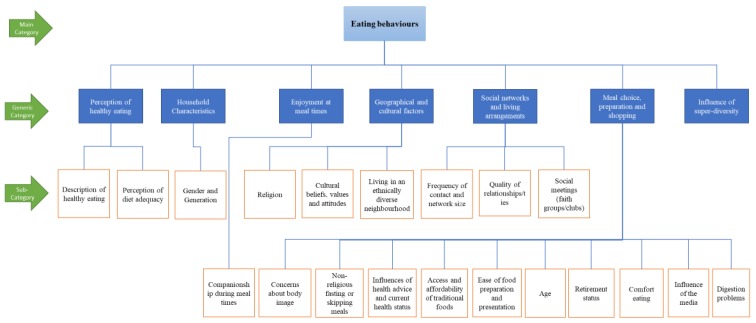
Complete coding matrix on eating behaviours.

**Figure 2 nutrients-11-01224-f002:**
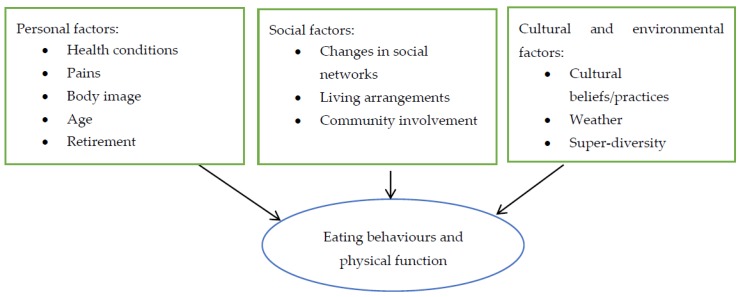
Influences on eating behaviours and physical function among community-dwelling ethnically diverse older adults.

**Table 1 nutrients-11-01224-t001:** Socio-demographic characteristics of participants completing interviews at baseline and 8-month follow-up.

Variable		Baseline			Follow-Up		
Sex N (%)		Male 57 (57.0)	Female 35 (35.0)	Total 92 (100)	Male 50 (62.0)	Female 32 (38.0)	Total 81 (100)
Age mean (SD)		71(7.0)	70 (9.0)	70.6 (8.1)	70 (8.0)	71 (9.0)	70.7 (8.2)
Ethnicity N (%)	Caribbean	14 (24.6)	25 (71.4)	39 (42.4)	13 (26.0)	25 (80.6)	38 (46.9)
	Pakistani	17 (29.8)	3 (8.6)	20 (22)	14 (28.0)	1 (3.2)	15 (18.5)
	African	11 (19.3)	6 (17.1)	17 (18.5)	11 (22.0)	4 (12.9)	15 (18.5)
	Bangladeshi	4 (7.0)	0 (0.0)	4 (4.3)	2 (4.0)	0 (0.0)	2 (2.5)
	Indian	5 (8.8)	1 (2.9)	6 (6.5)	4 (8.0)	1 (3.2)	5 (6.2)
	Others^#^	6 (10.5)	0 (0.0)	6 (6.5)	6 (12.0)	0 (0.0)	6 (7.4)
Marital status N (%)	Married	46 (80.7)	16 (45.7)	62 (67.4)	40 (80.0)	15 (48.4)	55 (67.9)
	Widowed	6 (10.5)	8 (22.9)	14 (15.2)	4 (8.0)	7 (22.6)	11 (13.6)
	Divorced	4 (7.0)	8 (22.9)	12 (13.0)	5 (10.0)	7 (22.6)	12 (14.8)
	Single	1 (1.8)	3 (8.6)	4 (4.3)	1 (2.0)	2 (6.5)	3 (3.7)
Faith/Religion N (%)	Christian	23 (40.4)	30 (85.7)	53 (57.6)	22 (44.0)	28 (90.3)	50 (61.7)
	Muslim	29 (50.9)	1 (2.9)	30 (32.6)	24 (48.0)	0 (0.0)	24 (29.6)
	Sikh	3 (5.3)	3 (8.6)	6 (6.5)	2 (4.0)	2 (6.5)	4 (4.9)
	Hindu	2 (3.5)	0 (0.0)	2 (2.2)	2 (4.0)	0 (0.0)	2 (2.5)
	No religion	0 (0.0)	1 (2.9)	1 (1.1)	0 (0.0)	1 (3.2)	1 (1.2)
Education N (%)	College/University	28 (49.1)	17 (48.6)	45 (48.9)	26 (52.0)	15 (48.4)	41 (50.6)
	Secondary School	10 (17.5)	8 (22.9)	18 (19.6)	8 (16.0)	7 (22.6)	15 (18.5)
	Primary School	9 (15.8)	6 (17.1)	15 (16.3)	9 (18.0)	5 (16.1)	14 (17.3)
	No education	10 (17.5)	4 (11.4)	14 (15.2)	7 (14.0)	4 (12.9)	11 (13.6)
BMI categories N (%)	Normal	4 (7.0)	2 (5.7)	6 (6.5)	6 (12.0)	3 (9.7)	9 (11.1)
	Overweight	22 (38.6)	7 (20.0)	29 (34.5)	13 (26.0)	8 (25.8)	21 (25.9)
	Obese	31 (54.4)	26 (74.3)	57 (62.0)	31 (62.0)	20 (64.5)	51 (63.0)
IMD Quartile * N (%)	1 (Most deprived)	20 (35.1)	12 (34.3)	32 (34.8)	18 (36.0)	11 (35.5)	29 (35.8)
	2	13 (22.8)	6 (17.1)	19 (20.7)	13 (26.0)	6 (19.4)	19 (23.5)
	3	11 (19.3)	6 (17.1)	17 (18.5)	6 (12.0)	5 (16.1)	11 (13.6)
	4 (least deprived)	13 (22.8)	11 (31.4)	24 (26.1)	13 (26.0)	9 (29.0)	22 (27.2)
Self-rated health N (%)	Excellent	9 (15.8)	9 (25.7)	18 (19.6)	8 (16.0)	8 (25.8)	16 (19.8)
	Good	35 (61.4)	17 (48.6)	52 (56.5)	31 (62.0)	14 (45.2)	45 (55.6)
	Fair	7 (12.3)	6 (17.1)	13 (14.1)	5 (10.0)	6 (19.4)	11 (13.6)
	Poor	6 (10.5)	3 (8.6)	9 (9.8)	6 (12.6)	3 (9.7)	9 (11.1)
Length of stay in the UK mean (SD) (years)		41.9 (16.2)	43.8 (14.9)	42.71 (15.9)	41.5 (16.7)	45.9 (15.5)	43.2 (16.4)

BMI = Body Mass Index; IMD = Index of Multiple Deprivation; SD = Standard Deviation; ^#^Other ethnicity referring to East African Asians or Mixed ethnic minority group such as a Caribbean Asian; * WHO guidance on BMI thresholds for Asian populations (World Health Organization 2004) was used for categorizing BMI of South Asian participants and standard BMI categories were used for Caribbean and African participants.

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
