# Peer review of "Perceptions and Factors Influencing Eating Behaviours and Physical Function in Community-Dwelling Ethnically Diverse Older Adults: A Longitudinal Qualitative Study"

_nutrients, 2019, doi:10.3390/nu11061224_

Round 1
Reviewer 1 Report
The paper presents a qualitative longitudinal study of perceptions of factors of healthy eating and physical activity among of older individuals from diverse ethnic minorities living in an urban environment. Two interviews were conducted over an 8-month period. The study was informed by an ecological model including personal, social, environmental and cultural factors.
I found the study very interesting and addressing the growing need to understand factors of healthy lifestyle in ageing among ethnic minorities; this is yet an understudied topic and has very strong implications for health promotion. The paper is well-presented, with a clear rationale. The methodology is sound and the conclusions provided by the authors are well supported by the findings of the study.
I have some minor comments for the authors:
Abstract, line 20 - The abstract states 82 participants at follow-up, but the Results section describes 81 participants. Please resolve inconsistency.
Introduction, lines 40-51: It would be useful here, if possible, to provide a comparison between ethnic minorities and British national residents in relation to eating behaviours and physical activity. What is the percentage of UK residents meeting weekly physical activity recommendations? What is the prevalence of obesity? This would provide a clearer picture of the higher, if so, health risk for ethnic minorities than national residents.
Theoretical framework (section 2.3 and supplementary File 2) - The ecological model is based on Bronfenbrenner's model, thus, I would recommend adding the crono-system (changes over time) proposed in the original model so to better contextualise the follow-up interviews and category 3 in the results section.
Theoretical framework (section 2.3 and supplementary File 2) - I wonder why the personal level does not include psychological factors. This seems to emerge in the Results (e.g., body image).
Data analysis, lines 174-175 - Please clarify whether discussions were resolved by consensus or a third researcher was involved.
Results, lines 188-199: As ethnicity is central to this study, I would recommend adding a brief comment on the distribution of ethnicity groups in the study and also clarify what the "other" ethnicity group includes.
Discussion, section 4.2 - This section would benefit from some re-structuring and synthesising, as the discussion of personal, social and environmental levels appears to be fragmented (e.g., personal level mentioned at lines 698-699 then again at line 723, and similar patterns for social and environmental). Given the length of the paper, having a more succinct discussion would increase legibility.
Discussion, minor comment on limitations - An 8-month follow-up is more likely to be associated with short-term personal and social changes than environmental changes, thus a suggestion for future research would be to have investigations of this type over the longer term to see, for instance, the impact of increased urbanisation.
Author Response
Dear Editor and Reviewers,
Thank you so much for the opportunity to review our manuscript and for the timely feedback. Please find below responses to each of the comments, and the corresponding changes in the revised manuscript highlighted in red.
Reviewer 1.
The paper presents a qualitative longitudinal study of perceptions of factors of healthy eating and physical activity among of older individuals from diverse ethnic minorities living in an urban environment. Two interviews were conducted over an 8-month period. The study was informed by an ecological model including personal, social, environmental and cultural factors.
I found the study very interesting and addressing the growing need to understand factors of healthy lifestyle in ageing among ethnic minorities; this is yet an understudied topic and has very strong implications for health promotion. The paper is well-presented, with a clear rationale. The methodology is sound and the conclusions provided by the authors are well supported by the findings of the study.
Thank-you for the opportunity to revise the manuscript and for your positive feedback on this original longitudinal study exploring the perceptions of, and factors influencing eating behaviours and physical function among ethnic older minorities.
Minor comments:
Abstract, line 20 - The abstract states 82 participants at follow-up, but the Results section describes 81 participants. Please resolve inconsistency.
Response: Thank you for noticing this inconsistency. This inconsistency has now been resolved in the abstract, line 20 now reads “An in-depth interviewing approach was used at baseline (n=92) and after 8-months (n=81)”.
Introduction, lines 40-51: It would be useful here, if possible, to provide a comparison between ethnic minorities and British national residents in relation to eating behaviours and physical activity. What is the percentage of UK residents meeting weekly physical activity recommendations? What is the prevalence of obesity? This would provide a clearer picture of the higher, if so, health risk for ethnic minorities than national residents.
Response: We have now included in line 49-52 of a comparison of activity levels between ethnic minorities and the general population from the 2004 Health Survey England report. Now line 49-57 reads “Furthermore, the prevalence of low physical activity in the UK, defined as the participation in less than one 30-minute moderate/vigorous activity session a week on the average was observed to be high among ethnic minorities [6]. For instance, as compared with 32% in men and 37% women in the general population, low activity levels were found in 51% in both Bangladeshi and Pakistani men, and 68% in Bangladeshi and 52% in Pakistani women [6]. Together with the low physical activity levels, ethnic minorities have an increased prevalence of obesity as compared to the general population; 25% of Caribbean men are obese as compared to 23% in the general population, and the risk ratio of obesity in African women, Caribbean women, and Pakistani women is 2.0, 1.43 and 1.48 respectively, as compared with the general population [6]”
Theoretical framework (section 2.3 and supplementary File 2) - The ecological model is based on Bronfenbrenner's model, thus, I would recommend adding the crono-system (changes over time) proposed in the original model so to better contextualise the follow-up interviews and category 3 in the results section.
Response: Thank you, the chronosystem (personal, social and environmental changes over the study period) has now been added to our adapted ecological model located in supplementary file 2. Section 2.3 has also been revised, line 137-139 now reads “In addition to these stages, the Bronfenbrenner’s concept of chronosystem was added as the last level of the ecological model to conceptualise the changes, and interactions, of the different levels of influence over time”
Theoretical framework (section 2.3 and supplementary File 2) - I wonder why the personal level does not include psychological factors. This seems to emerge in the Results (e.g., body image).
Response: Thank you, this has now been added as examples of personal influences within our ecological model, it now reads in the ecological model in supplementary file 2 as “personal influences: Demographic, psychological factors, biological etc”.
Data analysis, lines 174-175 - Please clarify whether discussions were resolved by consensus or a third researcher was involved.
Response: All discussions were resolved by consensus between the lead researcher and two other researchers at separate meetings. This has now been clarified in line 178-180 to read “Afterwards, all discussions around the difficulty of use of the matrices, interpretations of the categories and the data coded were resolved by consensus, and the matrices further restructured. Following this step, the remaining transcripts were coded by EAA using the updated categorisation matrices”
Results, lines 188-199: As ethnicity is central to this study, I would recommend adding a brief comment on the distribution of ethnicity groups in the study and also clarify what the "other" ethnicity group includes.
Response: Thank you, a sentence has been added to line 195-197, which reads The majority of the sample comprised of Caribbean and Africans (60%), followed by South Asians (34%) and the smallest proportion were participants identifying as ‘others’: a mixture of the two major ethnic minority groups (6%) “ An additional comment has also been added as a footnote to table 1, line 208-209 which reads * Other ethnicity referring to East African Asians or Mixed ethnic minority group such as a Caribbean Asian.
Discussion, section 4.2 - This section would benefit from some re-structuring and synthesising, as the discussion of personal, social and environmental levels appears to be fragmented (e.g., personal level mentioned at lines 698-699 then again at line 723, and similar patterns for social and environmental). Given the length of the paper, having a more succinct discussion would increase legibility
Response: We appreciate the Reviewer’s comments and a need to increase the legibility of the discussion. The discussion was presented in a format to draw importance to the perceptions of healthy eating and physical function as separate entity from the personal, socio-cultural and environmental factors influencing these. For example, in line 687-690 highlights differing perceptions of healthiness and affordability of traditional foods. While 714 onwards describes the personal, social and environmental factors influencing eating behaviour and physical function. However, we acknowledge that this section of the discussion could be clarified and therefore, have rewritten some areas and removed some repetition. For example, line 747-748 “Contrary to another study, very few participants made changes to their diet [52]. In those that reported making changes, some made unhealthy changes such as skipping meals or replacing meal times with unhealthy snacks” was found to portray the same message as line 786-788 “Considering that most participants in this study had one or more diet-related non-communicable disease, it was surprising to note that at follow-up only a few participants reported making a positive change to improve their health condition(s).” Hence line 747-748 has now been taken out to improve on conciseness.
Discussion, minor comment on limitations - An 8-month follow-up is more likely to be associated with short-term personal and social changes than environmental changes, thus a suggestion for future research would be to have investigations of this type over the longer term to see, for instance, the impact of increased urbanisation.
Response: Thank you for your comment. We have now included added this to the strengths and limitations section of the study. Line 852-855 now reads “Additionally, eight months of follow-up maybe considered a relatively short period of time to observe particularly environmental changes, hence the authors recommend future studies incorporating a longer follow-up period”
Again, we thank the Editors and Reviewers for the positive and insightful responses to this manuscript. We have undertaken revisions based on your comments, and hope that it is now suitable for publication.
Yours sincerely,
Evans A. Asamane on behalf of the authors

Reviewer 2 Report
Mr. E. A. Asamane and colleagues carried out a longitudinal qualitative study regarding perceived eating behaviours and physical function in a relatively large sample of ethnically diverse older adults across the Birmingham area. They have demonstrated that perceptions of healthy eating differed by sex, ethnicity, age and educational attainment. Most participants rated their physical function highly and perceived that physical function was related to health, exercise, and the ability to carry out activities of daily living. Moreover, health conditions, pain, age and retirement were factors influencing eating behaviours and physical function. Also social networks have a crucial impact on health outcomes in ageing. This study is clear and well-designed, and it underlines the importance of taking into account the heterogeneities between and within ethnicities when health interventions are designed.
Author Response
Reviewer 2.
Mr. E. A. Asamane and colleagues carried out a longitudinal qualitative study regarding perceived eating behaviours and physical function in a relatively large sample of ethnically diverse older adults across the Birmingham area. They have demonstrated that perceptions of healthy eating differed by sex, ethnicity, age and educational attainment. Most participants rated their physical function highly and perceived that physical function was related to health, exercise, and the ability to carry out activities of daily living. Moreover, health conditions, pain, age and retirement were factors influencing eating behaviours and physical function. Also, social networks have a crucial impact on health outcomes in ageing. This study is clear and well-designed, and it underlines the importance of taking into account the heterogeneities between and within ethnicities when health interventions are designed.
Response: Thank you very much for your positive feedback and the opportunity to revise our manuscript.
Again, we thank the Editors and Reviewers for the positive and insightful responses to this manuscript. We have undertaken revisions based on your comments, and hope that it is now suitable for publication.
Yours sincerely,
Evans A. Asamane on behalf of the authors